# Faecal Microbiota Characterisation of *Potamochoerus porcus* Living in a Controlled Environment

**DOI:** 10.3390/microorganisms11061542

**Published:** 2023-06-09

**Authors:** Donatella Scarafile, Diana Luise, Vincenzo Motta, Caterina Spiezio, Monica Modesto, Marzia Mattia Porcu, Yadid Yitzhak, Federico Correa, Camillo Sandri, Paolo Trevisi, Paola Mattarelli

**Affiliations:** 1Department of Agricultural and Food Sciences, University of Bologna, Viale Fanin 44, 40127 Bologna, Italy; 2Department of Medical and Surgical Sciences, University of Bologna, Via Massarenti 9, 40138 Bologna, Italy; 3Department of Animal Health Care and Management, Parco Natura Viva-Garda Zoological Park, 37012 Bussolengo, Italy; 4Fondazione Bioparco di Roma, Viale del Giardino Zoologico, 00100 Rome, Italy

**Keywords:** gut microbiota, bifidobacteria, diet, *Potamocherus porcus*, beneficial microbes

## Abstract

Intestinal bacteria establish a specific relationship with the host animal, which causes the acquisition of gut microbiota with a unique composition classified as the enterotype. As the name suggests, the Red River Hog is a wild member of the pig family living in Africa, in particular through the West and Central African rainforest. To date, very few studies have analysed the gut microbiota of Red River Hogs (RRHs) both housed under controlled conditions and in wild habitats. This study analysed the intestinal microbiota and the distribution of *Bifidobacterium* species in five Red River Hog (RRH) individuals (four adults and one juvenile), hosted in two different modern zoological gardens (Parco Natura Viva, Verona, and Bioparco, Rome) with the aim of disentangling the possible effects of captive different lifestyle and host genetics. Faecal samples were collected and studied both for bifidobacterial counts and isolation by means of culture-dependent method and for total microbiota analysis through the high-quality sequences of the V3–V4 region of bacterial 16S rRNA. Results showed a host-specific bifidobacterial species distribution. Indeed, *B. boum* and *B. thermoacidophilum* were found only in Verona RRHs, whereas *B. porcinum* species were isolated only in Rome RRHs. These bifidobacterial species are also typical of pigs. Bifidobacterial counts were about 10^6^ CFU/g in faecal samples of all the individuals, with the only exception for the juvenile subject, showing 10^7^ CFU/g. As in human beings, in RRHs a higher count of bifidobacteria was also found in the young subject compared with adults. Furthermore, the microbiota of RRHs showed qualitative differences. Indeed, Firmicutes was found to be the dominant phylum in Verona RRHs whereas Bacteroidetes was the most represented in Roma RRHs. At order level, Oscillospirales and Spirochaetales were the most represented in Verona RRHs compared with Rome RRHs, where Bacteroidales dominated over the other taxa. Finally, at the family level, RRHs from the two sites showed the presence of the same families, but with different levels of abundance. Our results highlight that the intestinal microbiota seems to reflect the lifestyle (i.e., the diet), whereas age and host genetics are the driving factors for the bifidobacterial population.

## 1. Introduction

The gastrointestinal tract of most animals hosts a microbial community, known as the gut microbiota, which shapes their phenotypes by modulating a range of physiological processes [1]. The intestinal microbiota is, therefore, a set of host-specific microorganisms which have been selected through host-microbe interactions under phylogenetic evolution and transition of feeding behavior by the host [2]. Human microbiome research has been improving our understanding of the environment as a major driver of variability. Family studies show that genetic ancestry or individual polymorphic variants have a minor role in gut microbiome composition (<2%). In contrast, over 20% of the variance in microbiome diversity can be inferred from shared environmental factors, such as those associated with diet and lifestyle [3]. However, the origin and evolution of enterotypes in mammals remain yet to be elucidated due to a lack of information about ancestral enterotypes [4]. 

Because of their potential to modulate host health, the gut microbiomes of controlled environment animals have recently become an increasingly important area of research [5]. In a controlled environment, wild animals are subject to conditions that they would not experience in their natural habitat [6]. Research on animal microbiomes living in zoos suggests that a controlled environment generally reduces the alpha diversity of vertebrate skin and gut microbiomes, although the magnitude of these effects is variable [1]. A controlled environment also has effects on microbial beta diversity, although the difference is not always in the same direction [7]. The effects of captivity on the mammalian gut microbiome [6] showed up to 56% of the variation in gut microbiomes between a controlled environment and wild mammal populations, driven primarily by shifts in Firmicutes, Bacteroidetes, and Proteobacteria phyla. Similarly, a meta-analysis by Alberdi et al. (2021) showed that the gut microbiota of 24 vertebrate species exhibited shifts in microbial richness between controlled environments and wild populations. However, the specific mechanisms that drive these changes are variable and non-exclusive, and they can be grouped into the following five general categories: (1) dietary changes, (2) habitat homogeneity, (3) stress, (4) antibiotics, and (5) altered intraspecific interactions [7].

Feed supplements, such as probiotics—selective microbes that produce positive effects on both the microbiota and host—can modulate the gut microbiota and have been found to improve the health of individuals in a controlled environment [7]. In particular, for those individuals destined for reintroduction, probiotics inhibit the colonisation of pathogenic microbes through competitive exclusion and/or secondary metabolites, which can reverse dysbiosis within microbial communities [8,9].

*Bifidobacterium* spp. are naturally occurring residents within the gastrointestinal tracts of mammals and are usually considered beneficial [10,11]. Hence, it is important to understand the diversity of bifidobacteria in the gut microbiota. Due to their claimed health-promoting properties, *Bifidobacterium* spp. have been included in many formulations [12]. The potential mechanisms underlying the health benefits of *Bifidobacterium* include the suppression of gut pathogens’ growth, capability to alter gut metabolism and enhance epithelial barrier function, and the anti-inflammatory modulation of host immunity [13]. The presence and species of bifidobacteria can vary significantly across different animal species. Consequently, it is crucial to explore the diversity of bifidobacteria within the gut microbiota of animals in a controlled environment. Such research is significant because it not only could offer potential health benefits for the animal in a controlled environment but also could hold promise for applications in economically valuable animals [14]. Indeed, bifidobacteria, as an alternative to antibiotic growth promoters, is widely used in animals to provide them with the nutrients required for growth, to increase their growth performance and immune modulation [15,16].

Th pig is the ideal mammal model to reveal the origin and evolution of host-specific intestinal microbiota because its direct wild ancestor, wild boars (*Sus scrofa scrofa*), and close phylogenetic neighbors, Red River Hogs (RRHs) (*Potamochoerus porcus*), both species are available for comparison. However, modifying diet ingredients and introducing intensive farming based on ad libitum cereal and pulse feeding, caused changes in the composition of the intestinal microbiota originally found in the ancestors of pigs [2].

*P. porcus* (Linnaeus, 1758) is one of the two species of the genus *Potamochoerus* and among the smallest and most plesiomorphic (ancestral) of the eight African swine [17,18]. The RRHs generally prefer damp forests and they can be found in a variety of habitats throughout their distribution, but never far from thick vegetative cover, soft soils (for rooting), and water [18]. The RRHs feed on a great variety of food items, particularly tubers, and roots, which are uprooted with the snout, along with seeds gleaned from elephant dung and fruits, grass, aquatic plants, bulbs, fungi, and other seeds. Occasionally, they will also eat invertebrates, reptiles, eggs, young birds, and carrion. They feed on a wide range of cultivated plants, and in proximity to human settlements, they can cause severe damage to crops [18]. *Potamochoerus porcus* was first exhibited in a zoological setting in 1852 at the London Zoo, with successful breeding recorded there beginning in 1857 [18]. In human care, *P. porcus* is typically good-tempered and even “friendly” toward caregivers; some individuals will solicit tactile contact from keepers, although human-directed aggression was reported from a habituated male upon maturity. *P. porcus* is quite common in zoos. Due to its suitability to live in mixed-species exhibits, it has been successfully housed with a variety of taxa in captivity, including hoofstock, medium-sized primates, small carnivores, and various ground birds [18]. In Italy, RRHs can be found at Fondazione Bioparco in Roma (Rome) (two adult individuals) and at Parco Natura Viva (Bussolengo, Verona) (two adults and one young subject). The diet of RRHs in these two different zoological gardens shows a common basis with few differences. 

Therefore, to better understand the taxonomy of intestinal microbiota and the possible effects of husbandry practice of different facilities, the five Italian RRH individuals were analysed. In this study, the intestinal microbiota of RRHs were decoded to identify factors driving development and differences in their enterotypes due to different environments or diets. Furthermore, due to the importance of bifidobacteria as one of the most important beneficial groups in gut microbiota with potential application as probiotics, a culturomic approach aimed to individuate the RRHs bifidobacteria-dominated enterotype at the species level has been applied to the different subjects of *P. porcus*.

## 2. Materials and Methods

### 2.1. Animals and Sample Collection

In July 2018, fresh individual faecal samples from 5 subjects, 3 (2 adults and 1 young) housed under semi-natural conditions in Parco Natura Viva-Garda Zoological Park (Bussolengo, Verona, Italy) and 2 (adults) in Bioparco (Roma, Italy) were collected from the ground using a sterile spoon, put into a sterile plastic tube, and stored under anaerobic conditions in an anaerobic jar (Merck) at 4 °C. The anaerobic atmosphere was obtained using the GasPak EZ Anaerobic Pouch system (BD). 

Samples of faeces were collected by the animal-care staff (keepers) during their routine cleaning of the enclosure and were taken promptly to the laboratory (within 2 h). Animals were free from intestinal infections and did not receive antibiotics or probiotics for two months before samples were collected. 

The two diets consisted mostly of grass, fruits and vegetables, and protein-based feed. (For details, please see Appendix A).

### 2.2. Bifidobacterial Enumeration, Isolation, Genotyping, and Identification

For isolation and enumeration of the bifidobacteria, aliquots comprising approximately 1 g of the faecal sample were serially diluted (tenfold) with Peptone Water (Merck, Manama, Bahrain) supplemented with cysteine hydrochloride (0.5 g L^−1^). Aliquots of 1 mL from each dilution (from 10^−1^ down to 10^−9^) were inoculated onto modified MRS (mMRS) (Difco, Beirut, Lebanon) agar supplemented with mupirocin (100 mg L^−1^) (Applichem, Darmstadt, Germany). Plates were incubated in anaerobic conditions, at 37 °C for 48–72 h. The anaerobic atmosphere was obtained using the GasPak EZ Anaerobic Pouch system (BD). After incubation, morphologically different colonies were randomly picked and re-streaked for several generations to obtain purified individual bacterial isolates. Isolates were suspended in a 10% (*w*/*v*) sterile skim milk solution, supplied with lactose (3%, *w*/*v*) and yeast extract (0.3%, *w*/*v*), and kept both freeze-dried and frozen at −120 °C until further analysis. 

For species identification, isolated bifidobacterial strains were cultivated in MRS broth supplemented with cysteine hydrochloride (0.5 g L^−1^) and incubated in an anaerobic atmosphere at 37 °C for 16 h. Subsequently, cells were harvested by centrifugation at 3500× *g* for 8 min, and the obtained cell pellet was used for DNA extraction with the Wizard Genomic DNA kit (Promega, Alexandria, Australia), following the manufacturer’s instructions. 

The extracted DNA was then subjected to a genotyping PCR using the BOXA1R primer (5-CTACGGCAAGGCGACGCTGACG-3) and the conditions previously described [19] in order to identify only one representative strain per genotype. 

Furthermore, an identification protocol through a PCR-based methodology using genus-specific primers targeting the 16S rRNA gene and conditions, according to [20] was carried out. Amplified 16S rRNA partial gene sequences were purified and sequenced by Eurofins Genomics (München, Germany). Strains were identified by comparing their 16S rRNA partial sequence against EZBIOCLOUD (https://www.ezbiocloud.net; accessed on 1 March 2023) [21]. 

### 2.3. DNA Extraction and Sequencing

Total DNA was extracted from each faecal sample using the PowerSoil DNA Isolation Kit (Qiagen, Hilden, Germany) according to the manufacturer’s protocol. Extracted DNA was quantified with NanoDrop (Thermo Fisher Scientific, Wilmington, DE, USA) and diluted to 10 ng/μL with DNase- and RNase-free water. Libraries were constructed as follows. Briefly, the V3–V4 region of the bacterial 16S rRNA gene was amplified with universal primers (F: Pro341F: 5′-TCGTCGGCAGCGTCAGATGTGTATAAGAGACAGCCTACGGGNBGCASCAG-3′ and R: Pro805R: 5′-GTCTCGTGGGCTCGGAGATGTGTATAAGAGACAGGACTACNVGGGTATCTAATCC-3′) using PlatinumTM Taq DNA Polymerase High Fidelity (Thermo Fisher Scientific, Monza, Italy). The PCR reaction conditions for amplification of DNA were as follows: initial denaturation at 94 °C for 60″, followed by 25 cycles of denaturation at 94 °C for 30″, annealing at 55 °C for 30″ and extension at 65 °C for 45″, ending with 1 cycle at 68 °C for 7′. Agarose gel electrophoresis was performed to verify the presence and size of amplicons.

The total microbiota of all samples were then decoded through a next-generation sequencing (NGS) approach of the V3–V4 region of bacterial 16S rRNA using a MiSeq platform (Illumina, San Diego, CA, USA) at BMR Genomics (Padova, Italy). The libraries were prepared using the standard protocol for MiSeq Reagent Kit v3, and were sequenced on the MiSeq platform (Illumina Inc., San Diego, CA, USA). The raw sequences were processed using the DADA2 pipeline, and the Silva (release 138) database was used as reference for the taxonomy assignment. For the DADA2 pipeline, primers were removed from the raw sequences, and based on the average quality score, forward and reverse reads were trimmed at position 290 and 250. All other DADA2 parameters were left with their default settings. 

### 2.4. Statistical Analysis

The statistical analysis was carried out in an R v4.1 environment [22] using the PhyloSeq [23], Vegan [24] and Deseq [25] packages. The alpha diversity indices (Chao1, Shannon and InvSimpson) were calculated and analysed using an ANOVA model considering the sites (Rome or Verona) as fixed factors. For the beta diversity, a Non-metric Multi-dimensional Scaling (NMDS) plot using the Bray–Curtis distance matrix was created. The effect of location was tested using the Adonis function with 999 permutations, and the pairwise comparison was carried out using the pairwise Adonis function [26]. Prior to the Adonis test, the homogeneity of dispersion among the different locations and among ages was tested using the betadisper function. Variables were removed from the model when not significant. The differences in taxonomic abundances among the sites were analysed using the DESeq2 package, based on negative binomial generalised linear models, including and applying the Benjamini–Hochberg method for multiple testing correction (estimate SizeFactors function) [25]. The results were considered significant when *p* was <0.05, and tendencies were 0.05 < *p* < 0.10; a false discovery rate (FDR) <0.1 and an LDA score cutoff of two were used in order to distinguish the differentially abundant taxa.

## 3. Results

### 3.1. Bifidobacterial Enumeration, Isolation, Genotyping and Identification

Bifidobacteria were detected in all analysed samples of RRHs from different zoos (Table 1). Count results showed levels ranging from 10^6^ in adult RRHs to 10^7^ CFU/g in the juvenile RRH. In total, 43 *Bifidobacterium* isolates were obtained. 

Before a species-level identification by means of 16S RNA comparative analysis, a preliminary screening of all isolates fingerprinting profiles with BOX-PCR was carried out. Results showed that the faecal samples from Bioparco Roma contain only identical strains, whereas samples from Verona contain two different strains (Appendix A). Therefore, BOX-PCR typing allowed the identification of 3 clusters: representatives of each cluster were selected and then subject to 16S rRNA sequencing. Comparative analysis of 16S rRNA partial gene sequences showed that strains from Roma RRHs had the highest similarity to *B. porcinum* (99.9%), whereas isolates from Verona RRHs had the highest similarity to *B. boum* (99.9%) and *B. thermoacidophilum* (99.9%). The occurrence of isolates in different samples is shown in Table 2.

### 3.2. Microbiota Composition of RRHs

A total of 334,035 good-quality reads were obtained from the five faecal samples. The relative rarefaction curves are reported in Figure 1. The tendency to a plateau for the curves of each sample suggested that the sequencing depth was sufficient for describing the variability within the microbial communities analysed.

The DADA2 pipeline identified a total of 1204 unique ASVs, from which a total of 14 different phyla (the most abundant were: 49% Firmicutes, 38% Bacteroidota, 8% Spirochaetota), 41 orders (Bacteroidales 38%, Oscillospirales 29%, Lachnospirales 9% and Spirochaetales 8%) 64 families (18% Oscillospiraceae, 15% Rikenellaceae, 9% Lachnospiraceae and 8% Spirochaetaceae), and 128 genera (most abundant were: 21% Rikenellaceae_RC9_gut_group, 11% UCG-005, 10% *Treponema* and 8% Christensenellaceae_R-7_group) were identified among the samples. The relative abundance of the 30 most abundant ASVs, at the phylum, order, family, and genus level, is shown in Figure 2. 

Results for alpha diversity, defined as the average species diversity within samples, are reported in Figure 3. 

The site of sampling tended to influence the Chao1 (*p* = 0.075), but significantly influenced the Shannon and InvSimpson diversity indices (*p* = 0.025; *p* < 0.01). Rome RRH samples showed a trend for a higher Chao1 index when compared with the Verona RRH samples (Roma = 530, Verona = 466, *p* = 0.07). However, the Verona RRH samples revealed higher Shannon and InvSipmson indices when compared with the Roma RRH samples (Shannon:Roma = 4.69; Verona = 4.92, *p* = 0.025; InvSimpson: Roma = 27.2; Verona = 62.3). Regarding the beta diversity, Figure 4 shows the NMDS plot using the Bray–Curtis distance matrix; Rome and Verona RRHs are separated into two distinct clusters. It is interesting to note that sampling sites tended to influence the richness (Chao1, *p* = 0.075) that is higher in Rome than in Verona. On the contrary, evenness (Shannon: *p* = 0.025) and diversity (InvSimpson: *p* = 0.05) were significantly higher in Verona than in Rome. These alpha indices have a different mode of calculation [27,28]; having a higher Chao index and lower Shannon and InvSimpson indices indicate a high richness (especially with rare taxa), but an uneven Rome site compared with the Verona site. Regarding the beta diversity, Figure 4 shows the NMDS plot using the Bray–Curtis distance matrix; Rome and Verona RRHs are separated into two distinct clusters.

Results for the different taxa in the two sites of sampling at family and genera levels are reported in Table 3. 

At the family level, the Rome site had a higher abundance of Oscillospirales_UCG-011, Paracaedibacteraceae, Sutterellaceae, and Paludibacteraceae when compared with the Verona site (*p* < 0.01), which in turn showed a higher relative abundance of Spirochaetaceae, Akkermansiaceae, Defluviitaleaceae, p-2534-18B5_gut_group, Lactobacillaceae, and Corynebacteriaceae (*p* < 0.05). At the genus level, the Rome Site had a higher abundance of Lachnospiraceae_UCG-003, *Cellulosilyticum*, Prevotellaceae_UCG-004, *Lachnospira*, *Alloprevotella,* and *Sutterella* compared with the Verona site (*p* < 0.05). In comparison, the Verona site had a higher relative abundance of *Treponema*, *Blautia*, *Candidatus*, *Soleaferrea*, Erysipelotrichaceae_UCG-003, *Anaerostipes*, *Dorea*, *Papillibacter*, Lachnospiraceae_XPB1014_group, *Lactobacillus*, Rikenellaceae_dgA-11_gut_group, and *Corynebacterium* (*p* < 0.05). 

*Bifidobacterium* relative abundance in the analysed samples showed a higher abundance only in the juvenile RRH Verona subject (Figure 5). 

## 4. Discussion

Commensal microorganisms are essential for the normal development and function of many aspects of animal biology, including digestion, nutrient absorption, immunological development, behaviors, and evolution [29]. The establishment of the mammalian gut microbiota starts upon birth, as the initially sterile gastrointestinal tract of the newborn becomes populated by bacteria originating from the mother or present in the surrounding environment [30]. 

Humans, especially breastfed infants, harbour *Bifidobacterium* as a predominant lactic acid bacteria [31], which may explain their low susceptibility to enteric diseases [32]. Contrarily, in the development of pig intestinal microbiota, there is the predominance of *Lactobacillus* spp., considered a key event for health promotion [33,34], due to the link with the significant reduction in *Enterobacteriaceae* and *Clostridium perfringens* [35]. Such development is affected by weaning, dietary changes, and host immune development [36,37,38,39]. Together with *Lactobacillus* spp., bifidobacteria are also among the first gut colonisers of newborns of various animals (including pig), that provide parental care to their offspring [14,40,41,42]. Moreover, bifidobacteria have been shown to have crucial roles in a variety of biological processes, such as the development of the gastrointestinal tract, the induction of mucus layer production, protection against pathogens, maturation of the immune system, as well as expansion of the gut glycobiome and participation in the processing of indigestible food components [43,44,45].

With the aim to compare different rearing conditions, 16S rRNA V3-V4 regions from the five faecal samples obtained from *Potamochoerus porcus* in Rome and Verona were sequenced to detect their microbiome composition. The results show good sequencing efficiency and highlight some aspects that allow discrimination between the two considered groups (Rome RRHs vs. Verona RRHs). The relatedness of each metagenomic dataset showed two distinctive clusters, a cluster composed of RRHs in Rome with a greater frequency of Bacteroidetes and one composed of RRHs in Verona with a greater frequency of Firmicutes (Appendix A). Furthermore, the two clusters would be driven by the differences in taxa diversity indices and the abundance of specific taxa, including Lachnospiraceae_UCG-003, Prevotellaceae_UCG-004, and *Alloprevotella* which were more abundant in the Rome site, and *Blautia*, Erysipelotrichaceae_UCG-003, *Dorea*, Lachnospiraceae_XPB1014_group, and *Lactobacillus* which were more abundant in the Verona site. All the mentioned taxa are considered normal and common taxa in the commensal microbiomes of domestic pigs [46,47].

Diet is considered one of the main drivers in the modulation of gut microbiota [48]. The so-called “Western diet,” characterised by a higher intake of simple sugars, fats, and animal proteins, is associated with a microbiota characterised by a greater presence of *Firmicutes*, in particular, *Clostridiales*, to the detriment of *Bacteroidetes*, and is associated instead with diets with a higher intake of fiber [49,50,51]. A study of the faecal microbiota in *Potamochoerus larvatus* from a wild population of Madagascar [52] showed a greater frequency of Firmicutes than Bacteroidetes in this population, suggesting a diet rich in animal proteins. In our study, even though the two diets (Rome and Verona) consist mostly of grass, fruit, vegetable, and protein-based feed, as reported in Appendix A, these kinds of food were balanced differently in the two parks, with the results being that the number of fruits per animal compared against vegetables was higher in Parco Natura Viva, Verona, than in Bioparco of Rome. In addition, plant proteins were higher in the diet examined in Rome than in the diet examined in Verona. Indeed, in Rome, alfalfa hay, which is known for its protein value [53] was used instead of hay. In Parco Natura Viva, on the other hand, the diet had more animal protein with dog pellets than the diet in Bioparco of Rome, where pig pellets were used [53]. The differences in the diets with more animal proteins in Verona and in plant proteins in Rome could be linked to the prevalence in the gut microbiota of RRHs of Firmicutes in Verona and of Bacteroidetes in Rome. 

Concerning geographical conditions, the Bioparco of Rome is located in the city, whereas Parco Natura Viva (Verona) is spread over 42 acres of Morainic woodland. Indeed, Rome has different weather from Parco Natura Viva, Verona, with more sun and higher temperatures all through the year. Additionally, the geographical location could be considered an environmental driver of the modulation of microbiota, even if with a minor impact with respect to diet [2].

Considering the importance of bifidobacteria for the host health and the difficulties sometimes in evidencing their presence by NGS [54,55], in this study, culturomic analyses have been used, together with the NGS approach, to study the presence of the bifidobacterial population. Using culturomics, the bifidobacterial levels in fecal samples of *P. porcus* have been clearly evidenced, showing counts ranging from 10^6^ CFU/g to 10^7^ CFU/g. As in human beings, in RRHs higher counts of bifidobacteria were also found in the young subject compared with the adults [11]. This result was also confirmed by NGS, but only in the juvenile subject (RRHc in Verona) (Figure 5).

With culturomics, species of bifidobacteria reported in the literature in pigs are Bifidobacterium longum subsp. suis, B. pseudolongum subsp. globosum, B. pseudolongum subsp. pseudolongum, B. thermophilum, *B. thermoacidophilum*, *B. boum*, B. choerinum and *B. porcinum* [56]. In the present study, different bifidobacterial species distribution was found with respect to the site of sampling. Indeed, RRHs in Rome showed only the presence of *B. porcinum*, whereas Verona RRHs had *B. boum* and *B. thermoacidophilum* (Table 2). Culturomics presents a noteworthy benefit: it provides access to strains that can serve as probiotics for various applications. Animals housed in zoos, for example, can potentially benefit from specific probiotics such as bifidobacteria, which have been isolated from them. These probiotics can be administered to the same animals during periods of gut microbiota imbalance caused by stressors. By doing so, a healthy equilibrium of beneficial bacteria in the gastrointestinal tract can be maintained, supporting optimal digestion and nutrient absorption, and boosting the immune response to combat diseases and infections.

Anatomical modifications of the intestine do not seem to have much impact on the intestinal microbiota because the enterotype in wild boars and wild *Potamochoerus* closely resembles that in domestic pigs raised on artificial formula feeding [2].

The specific microbial composition of the intestinal tracts of RRHs remains poorly characterised in the literature. Therefore, this study aimed to evaluate the composition, distribution, and development of the intestinal microbiome in RRHs raised under semi-natural conditions, despite the limitation of analysing a small sample size consisting of only five individuals. These five individuals, three located in Verona and two in Rome, represent the entirety of RRHs found in Italian zoos. Future studies with more subjects, both in zoos and in the wild, will improve and strengthen comparison between the microbiota of animals living in controlled environments and in nature, allowing us to verify if the diet adopted by different facilities, such as zoos, is able to preserve the species gut core microbiota described in the wild. In fact, comparing controlled environments and wild populations can help to identify “marker” taxa of the core microbiota. This should improve the husbandry and conservation of zoo animals, modulating their diet and making it as similar as possible to that reported in their wild counterpart. This parameter could also be useful to control if the food plans of different parks overlap. 

## 5. Conclusions

Zoo-housed wild animals have environments and lifestyles that they would not experience in their natural habitat. They cope with environment, diet, healthcare, and social interactions which are different from their wild counterparts. These circumstances can lead to improved animal welfare and longevity in some species, but for other species they can lead to health problems when under a controlled environment, including issues with metabolism and digestion. 

Very recently, the microbiome has been suggested to be useful as a mediator of host conditions. However, since drivers of microbiome changes under human care and also in the wild remain unclear, the use of microbial interventions to help animals is limited. Any study that can help to improve our knowledge on the microbiomes of different species and their drivers can be considered a very important step to reach the goal of improving the health of the animals by manipulating their gut microbiota. In addition, to characterise the gut microbiota of animals in a controlled environment, add samples, compare results with animals from the wild, and make the results available, though few animals are involved, is a starting point for further studies in order to improve the health and the welfare of animals in a controlled environment.

## Figures and Tables

**Figure 1 microorganisms-11-01542-f001:**
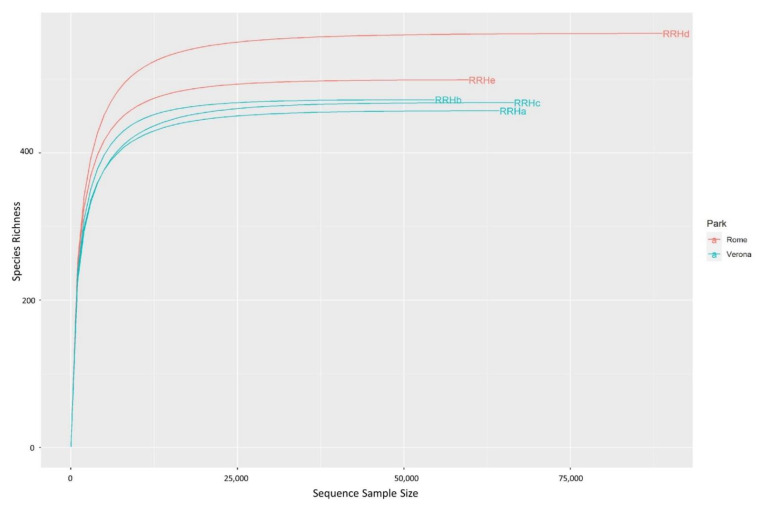
Rarefaction curves of the Red River hog faecal samples. Different colours have been used for the samples originating from each different zoo. Samples were sequenced for the the V3–V4 regions of bacterial 16S rRNA using a MiSeq platform (Illumina). Sequences were processed using the DADA2 pipeline, and were annotated using the Silva (release 138) database.

**Figure 2 microorganisms-11-01542-f002:**
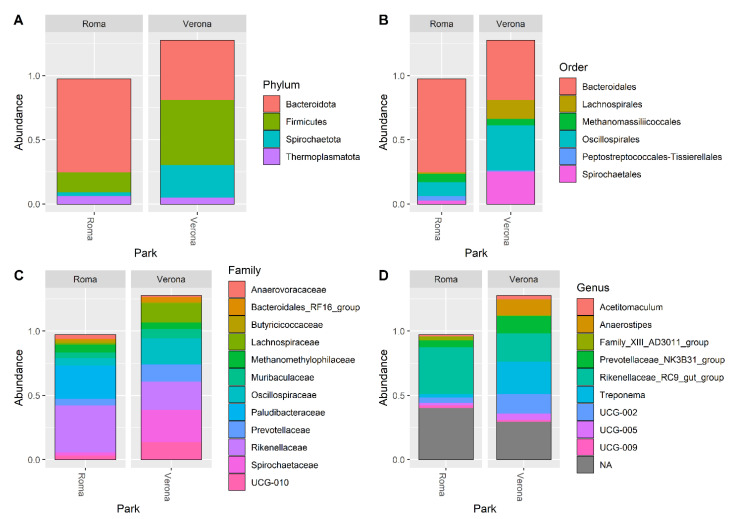
Bar plots representing the percentage abundance of the top 30 ASVs reported at Phylum (**A**), Order (**B**), Families (**C**), and Genera (**D**) levels.

**Figure 3 microorganisms-11-01542-f003:**
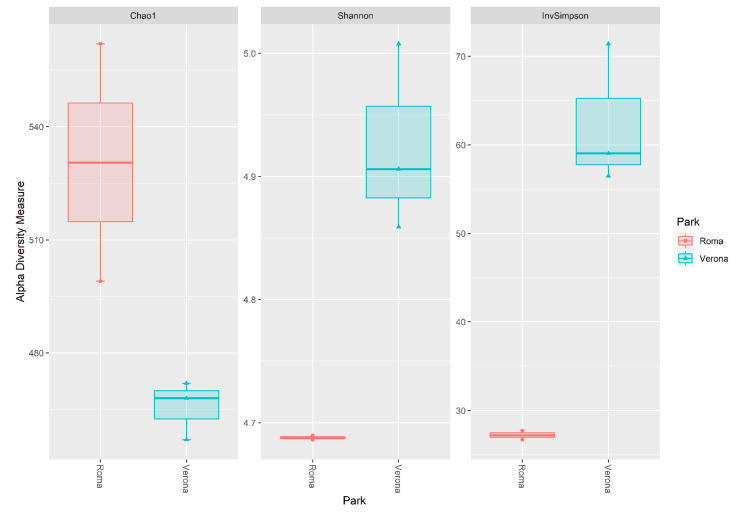
Box plot of the alpha diversity indices (Chao1, Shannon, invSimpson) estimated for the Rome and Verona sites.

**Figure 4 microorganisms-11-01542-f004:**
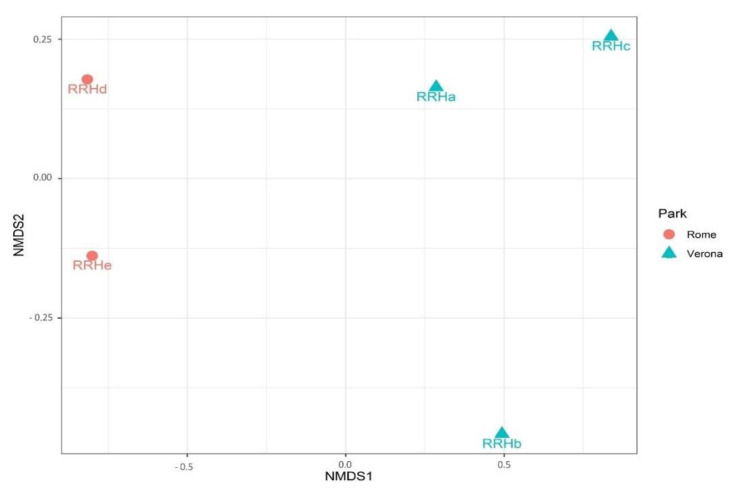
NMDS plot on Bray–Curtis distance matrix on faecal samples of Red River located in the different sites.

**Figure 5 microorganisms-11-01542-f005:**
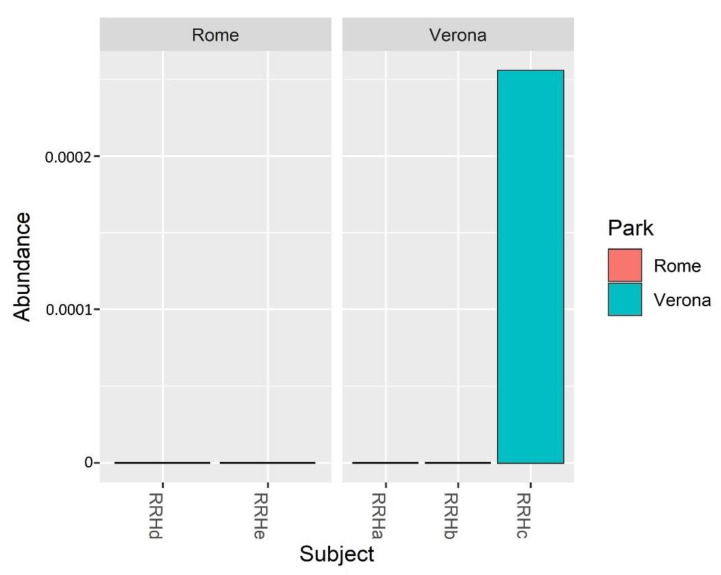
Bar plots showing the *Bifidobacterium* spp. relative abundance in the sampled subjects.

**Table 1 microorganisms-11-01542-t001:** Bifidobacterial counts enumerated in the five RRH faecal samples. Results are expressed as CFU/g of faecal sample.

Subject	Age	Sex	Born at	Living in	Counts
RRHa	Adult	male	Touroparc (Francia)	Parco Natura Viva, Verona (Italy)	2 × 10^6^
RRHb	Adult	female	Westfalischer (Germania)	Parco Natura Viva, Verona (Italy)	5 × 10^6^
RRHc	Juvenile	male	Bussolengo	Parco Natura Viva, Verona (Italy)	3 × 10^7^
RRHd	Adult	male	Wroclaw (Polonia)	Bioparco, Rome (Italy)	2 × 10^6^
RRHe	Adult	male	Wroclaw (Polonia)	Bioparco, Rome (Italy)	2 × 10^6^

**Table 2 microorganisms-11-01542-t002:** Occurrence of *Bifidobacterium* species in *P. porcus* in samples from Verona and from Rome.

	Number of Isolates				
Species	Verona			Rome	
	RRHa	RRHb	RRHc	RRHd	RRHe
*B. thermoacidophilum*	7	5	6	0	0
*B. boum*	2	4	2	0	0
*B. porcinum*	0	0	0	11	8

**Table 3 microorganisms-11-01542-t003:** Different taxa observed in the faecal samples of RRHs in the Rome and Verona parks.

Site	baseMean^1^	log2FoldChange^2^	lfcSE^3^	Padj^4^	Taxa Name
Family level					
Rome	32.57	−8.58	2.04	0.000	Oscillospirales UCG-011
Rome	19.98	−7.87	2.32	0.005	Paracaedibacteraceae
Rome	77.48	−4.30	0.79	0.000	Sutterellaceae
Rome	4014.90	−3.95	0.82	0.000	Paludibacteraceae
Verona	6104.15	2.02	0.59	0.005	Spirochaetaceae
Verona	163.14	2.76	0.90	0.016	Akkermansiaceae
Verona	43.72	3.32	0.95	0.005	Defluviitaleaceae
Verona	294.27	7.19	2.62	0.039	Bacteroidales_p-2534-18B5_gut_group
Verona	210.07	11.07	1.68	0.000	Lactobacillaceae
Verona	21.74	20.63	4.41	0.000	Corynebacteriaceae
Genus level					
Rome	17.53	−7.98	2.28	0.01	Lachnospiraceae_UCG-003
Rome	10.96	−7.30	2.66	0.04	*Cellulosilyticum*
Rome	59.64	−5.80	1.68	0.01	Prevotellaceae_UCG-004
Rome	39.19	−5.18	1.84	0.04	*Lachnospira*
Rome	346.37	−4.81	0.67	0.00	*Alloprevotella*
Rome	86.85	−4.73	0.91	0.00	*Sutterella*
Verona	4744.10	2.06	0.73	0.04	*Treponema*
Verona	275.20	3.36	0.97	0.01	*Blautia*
Verona	249.39	3.40	0.95	0.01	Candidatus_Soleaferrea
Verona	37.42	5.14	1.71	0.02	Erysipelotrichaceae_UCG-003
Verona	1645.90	6.34	0.94	0.00	*Anaerostipes*
Verona	11.92	6.73	2.41	0.04	*Dorea*
Verona	14.96	7.06	2.34	0.02	Papillibacter
Verona	67.53	9.23	1.78	0.00	Lachnospiraceae_XPB1014_group
Verona	174.23	10.60	1.72	0.00	*Lactobacillus*
Verona	239.91	11.06	1.74	0.00	Rikenellaceae_dgA-11_gut_group
Verona	20.93	20.36	3.63	0.00	*Corynebacterium*

baseMean^1^ = mean of normalised taxa counts averaged over all samples from both conditions. log2FoldChange^2^ = log2 Fold Change. The sign is relative to the Rome group. lfcSE^3^ = log2 Fold change standard error. Padj^4^ = Benjamini–Hochberg adjusted *p* value.

## Data Availability

Raw sequence are publicly available at NCBI Sequence Read Archive (SRA) under the accession number PRJNA951612.

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
