# Peer review of "Faecal Microbiota Characterisation of Potamochoerus porcus Living in a Controlled Environment"

_microorganisms, 2023, doi:10.3390/microorganisms11061542_

Round 1
Reviewer 1 Report
The manuscript characterized the faecal microbiota of Red River Hog in different controlled environments with an emphasis on Bifidobacteria. Most of the data provided in the manuscript are reliable and the result is technically sound. Nevertheless, there are concerns.
1. The study mainly focused on the Bifidobacterium in Potamochoerus porcus, please explain why it is relevant and please provide further evidence from the 16S rRNA data on it.
2. The introduction and discussion part have too many paragraphs and lacks coherence. It is recommended to reorganize.
3. The conclusion of the paper is that “Our results highlight that the intestinal microbiota seems to reflect the lifestyle (i.e., the diet), while the age and host genetics are the driving factors for the bifidobacterial population.” Please provide the diet, lifestyle and environmental differences between the two zoos.
4. Line 13-16, The font is inconsistent with the rest of the manuscript.
5. The right side of Figure 1 is incomplete, and the annotation information of microbiota samples is missing.
6. The units in Section 2.2 seem to be wrong.
7. Both Line166 and Line227 provide the abbreviated information of NMDS, which is usually based on the first occurrence.
8. Line148-150, please provide the specific min or sec instead of ' or ''.
9. Line 193-194, the result of B. boum, B. thermoacidophilum and B. porcinum was not shown in the manuscript.
Author Response
Please find our response in the attached rebuttal letter.
Best Regard.
Monica Modesto

Reviewer 2 Report
The topic of the work is important in any case, since it is particularly important in zoos to have parameters that allow animal welfare to be assessed and optimized.
However, I wonder whether it is really possible and useful to derive such things if only 2-3 animals were analyzed per group to be examined. The authors address this condition, but do not provide any information on how they will improve it. I'm also missing a clear take home message in the Conclusion chapter.In summary, I am not convinced that this work can really be of added value for the Potamochoerus porcus species kept in zoologic gardens.
Author Response

(The authors gave the same response as above.)
